## Research Article

forced migration; migrants; mental wellbeing; self-reliance and housing

**Corresponding author:**
Ilana Seff;
Email: seff@wustl.edu

# Beyond shelter: Exploring the potential impacts of rental assistance on self-reliance and well-being for Venezuelan migrants in Colombia

Lindsay Stark[1] [ID], Juan Pablo Franco[2], Arturo Harker Roa[3,4], Neema Mosha[5,6], Deanna Barch[1], Ned Meerdink[7] and Ilana Seff[1] [ID]

[1]Washington University in St. Louis, USA; [2]Blumont, USA; [3]School of Government, Universidad de los Andes, Colombia; [4]Imagina Research Center, Universidad de los Andes, Colombia; [5]Institute for Medical Information Processing, Biometry, and Epidemiology, Ludwig Maximilian University, Germany; [6]Mwanza Intervention Trials Unit, Tanzania, United Republic of and [7]Refugee Self-Reliance Initiative, USA

## Abstract

Urban refugees in low- and middle-income countries (LMICs) often face housing insecurity, undermining their ability to achieve self-reliance and well-being. Few studies have evaluated the impact of housing interventions in these contexts. This study offers preliminary evidence on the effectiveness of a 9-month rental assistance program targeting female-headed Venezuelan migrant households in Colombia. Using pre-post data from 517 participants, we assessed changes over time in household-level self-reliance, domains of self-reliance, subjective well-being and perceived agency. We also employed ordinary least squares regression and fixed-effects models to estimate changes in self-reliance and the relationship between self-reliance, psychosocial and housing outcomes. Our analysis found significant improvements in overall self-reliance, well-being and agency after controlling for observed individual and household characteristics. Increases were observed across almost all domains of self-reliance. Fixed-effects models also found that subjective well-being, perceived agency and select housing conditions were positively associated with self-reliance. Rental support appears to promote both material and psychosocial recovery for displaced households by alleviating financial stress and enabling forward-looking behaviors. However, the impact of housing quality dimensions varies, and the sustainability of outcomes remains uncertain. Future evaluations should incorporate longitudinal designs and control groups to inform holistic refugee housing strategies.

## Impact statement

Despite its recognized role as a barrier to refugee self-reliance and well-being in non-camp settings, housing insecurity has received limited empirical attention – particularly in urban areas of low- and middle-income countries (LMICs). This study provides preliminary evidence showing that a 9-month rental assistance program targeting female-headed Venezuelan migrant households in Colombia is significantly associated with improvements in self-reliance, subjective well-being and perceived agency. It also highlights that certain housing quality factors – such as safety and protection from the elements – are more closely linked to self-reliance gains than others. Findings from this study underscore the potential for housing subsidies to support both material and psychosocial recovery among displaced populations. The results also point to the need for more robust evaluations of housing-related interventions – particularly by including control groups and longer follow-up periods – to inform sustainable, scalable refugee housing strategies in LMIC settings.

## Introduction

Amid a growing prevalence of protracted and cyclical conflicts over the past several decades, global forced displacement has reached an all-time high (Blair et al., 2022). As of 2023, there were 50.3 million refugees, asylum seekers and other forced migrants in need of international protection living outside their country of origin (UNHCR, 2024). Refugees grapple with a range of negative exposures that emerge before, during and after migration, including experiences of violence and torture, loss of loved ones, the hardships and protection risks associated with forced displacement and the stressors that come with resettling in a new context, among many others (Gleeson et al., 2020; Mesa-Vieira et al., 2022). A substantive body of research has documented the implications of these cumulative risk factors on the mental health of refugees, with studies showing elevated levels of depression, anxiety and PTSD among this population (Bogic et al., 2015; Li et al., 2016; Rochlin, 2023). Evidence also suggests that certain mental health sequelae,

such as depression, may be most closely linked to challenging post-migration circumstances (Bogic et al., 2015).

Chronic poverty in the post-migration context, driven and exacerbated by a range of factors, including barriers to services and income-generation opportunities, has been widely documented as a predictor of anxiety, depression and other forms of psychological distress (Santiago et al., 2011). While the inability to meet one's basic needs is stressful in and of itself, the Scarcity Mindset framework offers insight into how persistent material insecurity may impact multiple dimensions of well-being. The Scarcity Mindset Framework posits that chronic financial insecurity – or even the *subjective belief* of financial scarcity – depletes cognitive bandwidth and increases cognitive tunneling, or the neglect of other needs (de Bruijn and Antonides, 2022). As a result, individuals struggle to engage in long-term planning and goal-oriented behaviors that are in the best interest of their long-term well-being (Haushofer and Fehr, 2014). In other words, the stress of poverty forces individuals to focus on immediate survival and basic needs, often at the expense of broader well-being, including accessing basic services, making social connections and fostering healthy behaviors (Rana et al., 2022). For example, one recent study found that a scarcity mindset was associated with reduced future-oriented decision-making around reproductive health among low-income women in Malawi (Norris et al., 2019).

Housing insecurity, which may serve as an outcome, co-condition and/or predictor of poverty, is a common concern among refugees. Securing safe and adequate housing is particularly relevant for refugees living in urban areas, which now account for 60% of refugees globally (Park, 2016). Housing insecurity has been widely evidenced as a social determinant of health, including mental health and emotional well-being (Mwoka et al., 2021). Further, the chronic stress of unstable housing can extend beyond individual mental health, affecting entire households by exacerbating financial insecurity, household distress and adults' use of unhealthy parenting practices (Warren and Font, 2015; Roberts et al., 2025). Evidence from certain low-income populations suggests that rental assistance programs can help alleviate financial strain and improve mental health outcomes among recipients (Fenelon et al., 2017). For example, low-income individuals receiving housing subsidies in the United States reported lower levels of psychological distress compared to their counterparts on waiting lists for this assistance (Denary et al., 2021). Among refugees, housing affordability, adequacy and insecure tenure have all been linked to increased psychological distress, anxiety and depression (Ziersch and Due, 2018). However, there is little evidence specifically on the impact of housing for refugee populations outside of camp settings or for those living in low- and middle-income country (LMIC) contexts (Brown et al., 2024).

Recognizing the impacts of housing insecurity and stress on income generation, it follows that an inability to secure safe and adequate housing may also hinder a household's path to self-reliance, defined as the ability to meet essential needs in a sustainable and dignified manner without external aid (Leeson et al., 2020). Securing stable housing may alleviate some of the cognitive burden associated with financial insecurity, freeing up mental space for securing a job, saving money and other behaviors that would promote sustainable self-reliance. Prior research has also demonstrated a link between self-reliance and mental health among forcibly displaced women, suggesting that housing security may have the potential to improve mental health and self-reliance in tandem or through mediating pathways (Seff et al., 2025). However, gaps remain in the literature regarding the impact of housing

support on self-reliance and broader psychosocial outcomes, particularly for refugees in low- and middle-income country (LMIC) contexts.

Housing insecurity in urban areas is of particular concern for Venezuelan refugees and migrants in Colombia. Home to 5.8 million refugees and migrants from Venezuela, Colombia hosts the most Venezuelan migrants in the region (Wolf, 2021; UNHCR, 2024). While Colombia has long grappled with internal displacement, managing large-scale transnational migration is a relatively new challenge, placing additional strain on an already overburdened system (Fernández-Nino and Bojorquez-Chapela, 2018; Aldana and Esteban, 2020). Refugee and migrant Venezuelan populations in Colombia face significant economic and social vulnerabilities, including barriers to employment, social protection, and stable housing (Correa-Salazar et al., 2025). Unsurprisingly, Venezuelan refugees and migrants in Colombia exhibit elevated levels of depression, anxiety and other mental health sequelae (Alarcon et al., 2022). Female-headed households in this context are particularly at risk, as they often rely on precarious employment in the informal sector, leaving them with limited financial security and heightened exposure to poverty and protection concerns (Jeronimo Kersh, 2021).

In Colombia, Venezuelan refugees and migrants often grapple with insecure housing due to and/or alongside financial insecurity. Although there is limited research on housing for refugees in urban areas of LMICs, existing evidence suggests renting is the most common housing arrangement for this population (Lombard et al., 2021); in urban Colombia, rentals account for 39% of housing (Blanco et al., 2016). Despite the availability of housing in urban areas, several barriers hinder refugees and migrants' ability to access adequate shelter, including cost, informal rental agreements leaving households vulnerable to eviction and discrimination (Scaramutti et al., 2024). In a recent assessment conducted by the Interagency Group for Mixed Migration Flows, shelter was the third most referenced need among Venezuelan migrants (72%), after food (86%) and jobs (77%) (Grupo Interagencial de Flujos Migratorios Mixtos Colombia, 2023). Even when households can secure housing, their shelters are often unsafe, lack basic sanitation and water facilities or are overcrowded. Despite these challenges, only 36% of households in the assessment reported receiving shelter assistance (Grupo Interagencial de Flujos Migratorios Mixtos Colombia, 2023).

A recently implemented rental subsidy program in Colombia offers an opportunity to expand the negligible evidence base on the impact of housing security on well-being and self-reliance among refugees. The rental assistance is embedded in *Acogida*, a larger program targeting female-headed Venezuelan refugee and migrant households. The program helps households identify safe and adequate housing and then covers the cost of rent for 9 months. Drawing on the Scarcity Mindset Framework, the intervention's theory of change posits that rental coverage serves as material support, which can directly reduce stress and anxiety for low-income households, and enables greater cognitive bandwidth, which in turn allows for future-oriented investment in stability and well-being. Together, these mechanistic pathways give rise to improved mental health and self-reliance. As part of the program's implementation, individual- and household-level data were collected from ~517 women before and after the provision of rental assistance. We employed fixed effects models and dynamic regression models to assess the program's association with changes in household-level self-reliance and subjective well-being. Findings offer theoretical and empirical contributions to help understand

whether and how housing assistance ensures more than safe and adequate housing, potentially fostering other outcomes around self-reliance, well-being and agency.

## Methods

This study presents an analysis of program data collected by Blumont, an international NGO, as part of their monitoring and evaluation efforts. Data were collected from intervention participants at both baseline and endline, and a deidentified data set was shared with the research team. This section presents details on the intervention, data collection procedures for the program data and analysis.

### Intervention

Families that were deemed eligible to participate in Blumont's rental support program were responsible for finding their own rental shelter (see below for inclusion criteria). They were required to identify a shelter that met three conditions: (i) the landlord needed to be willing to provide a written and signed rental agreement for 9 months; (ii) there could be no more than three people sleeping in one room; and (iii) the shelter needed to have access to water, sanitation and electricity. Blumont shelter officers then visited the rental properties and used a standardized checklist to verify compliance with these three conditions, as well as to ensure the home had adequate privacy and was not in a location at risk of natural disaster. Once rental conditions were confirmed, Blumont provided rental support for 9 months through four payments made directly to landlords. The first payment was provided immediately after the rental agreement was signed and covered the first 4 months of rent. Three additional payments were made every 2 months at the ends of months 4, 6 and 8. The average monthly rent covered as part of this intervention was COP$413,694 (USD$103).

Throughout the 9-month period of rental support, Blumont staff also made periodic visits to participants' homes to ensure continued compliance with the housing requirements and to make referrals, as needed, to social protection services. No direct services were provided by Blumont as part of this intervention; Blumont staff rather provided the names of relevant international and local partner organizations offering economic integration or protection services to participant families if requested.

### Participants and procedures

Blumont protection and monitoring and evaluation officers identified households eligible for participation in the intervention based on a predefined set of criteria, including having migrated from Venezuela, having a female household head and living in an inadequate shelter. Staff visited households living in 17 neighborhoods within 9 municipalities in October 2023. A scorecard was used to capture inclusion criteria, as well as to identify female household heads with additional vulnerabilities, including those who were survivors of gender-based violence, were involved in survivor sex, were pregnant or lactating, had a disability, were under 18 years or were elderly. Families with these vulnerabilities, as well as families in especially dire shelter situations (e.g., living in informal settlements, living in housing without ventilation or being at risk of eviction), received higher scores. Families with higher scores were then revisited in order to confirm they were interested in receiving rental support and could identify a rental solution that complied

with the conditions outlined above. Additional measures, not incorporated into the eligibility score, were also collected during this visit and are explained below.

Ultimately, 615 households met the inclusion criteria and were enrolled in the program. Additional data (primarily, the Self-Reliance Index [SRI]) were collected from these 615 households in January 2024, immediately after moving into the new and adequate shelter and before the first rental payment, and endline data were collected from 517 participants (16% were lost to follow-up) in August 2024, shortly before the end of the 9-month occupancy period. Data were collected by program staff, who received training on the purpose of the survey, the survey questions and data collection procedures, including consent. Informed verbal consent was obtained from all program participants.

### Measures

The primary outcome of interest, household-level self-reliance, was assessed using the SRI (Seff et al., 2021). The SRI conceptualizes self-reliance as a household's capacity to meet its essential social and economic needs sustainably. It encompasses 12 domains that capture a household's basic needs, available resources and capacity for long-term sustainability. SRI domains include housing adequacy, rent, food security, education, access to healthcare, health status, safety, employment, financial resources, assistance, debt, savings and social capital. Each domain captures either the household's current condition or status within the last 3 months. Importantly, because the baseline SRI data were collected immediately after households moved into their new housing, baseline housing scores were 5, on average, not leaving room for improvement at endline. Female heads of household responded to the SRI items on behalf of their entire household. The SRI was designed to be conversational in order to build rapport and capture a comprehensive assessment of the household's circumstances. As such, the tool utilizes guiding prompts for each domain rather than fixed, standardized questions. Final SRI scores and domain-specific scores range from 1 to 5, with higher scores indicating greater self-reliance. The SRI has been widely used at the global level and has been previously validated with Venezuelan migrants in Colombia (Seff et al., 2025).

Two key covariates of interest, subjective well-being and perceived agency, were measured in October 2023 and also included in the analysis. Subjective well-being was captured using the question: "Understanding well-being as the satisfaction you have with your life overall, indicate which step you are on today, where step 1 is the lowest level of well-being and step 5 is the highest level of well-being." Scores could assume a value from 1 to 5. Perceived agency was captured using the question: "If the current conditions of my life, allow me to act and/or make decisions about important objectives for my life project and that of my family." Respondents answered using a 5-point Likert scale from 1 "*Strongly disagree*" to 5 "*Strongly agree,*" with higher scores indicating greater perceived agency.

Other pre-post covariates explored included four indicators that captured various dimensions of living conditions, including having adequate space, safe shelter, required privacy and protection from elements or climate. Each dimension was presented as a statement (e.g., "The current housing has adequate space") and respondents answered on a 5-point Likert scale ranging from 1 "*Strongly disagree*" to 5 "*Strongly agree.*" Higher scores signal better living conditions. Unlike the measures above, which were taken in January 2024 immediately after households moved into their new shelters and at endline in August 2024, these living

condition variables were collected during the screening stage in October 2023 and again at endline in August 2024. Therefore, unlike the SRI housing domain, changes in these four housing indicators reflect changes from a period before any intervention to endline.

### Analysis

Data were transferred from Excel to STATA version 17 SE (StataCorp, Texas, USA) for cleaning and analysis. Descriptive statistics for continuous data were summarized using means and standard deviations, while categorical variables were summarized using frequency and percentages. T-tests were used to determine whether any key baseline characteristics differed between those who were measured both before and after the intervention, as compared to those who were lost to follow-up.

Program effects on the SRI domain scores, overall self-reliance and covariates of interest were assessed using bivariate ordinary least squares (OLS) regressions, where a dichotomous variable representing endline data collection served as the independent variable. These models also control for the respondent's age, the respondent's disability status and household size.

We then estimated the relationship between our main outcome of interest, self-reliance, the point of data collection and six covariates of interest: subjective well-being, perceived agency and the four housing indicators (equation 1). OLS regression models were used to estimate the change over time in SRI domain scores $(SRI_{it})$, accounting for variation in observed household-level characteristics measured at baseline $\left(\vec{X}_{it}\right)$. $SRI_{it}$ is the SRI score for household "*i*" in period "*t*," which ranges from 1 to 5 (higher = greater self-reliance). $endline_t$ is an indicator variable that equals 0 at baseline ($t = 0$) and 1 at endline ($t = 1$). Under this specification, $\beta_1$ quantifies the average change over time for the outcome variable, after accounting for the observed household characteristics.

$$SRI_{it} = \beta_0 + \beta_1 endline_t + \vec{\beta}_2 \vec{X}_{it} + u_{it} \quad (1)$$

To control for additional unobserved time-invariant characteristics that may also be correlated with these independent variables $(A_i)$, fixed-effect models ("xtreg, fe" in Stata) were also fitted to determine the association between the SRI score and each covariate (equation 2). Random-effect models, whereby we assume that time-invariant characteristics are not correlated with these independent variables, were also estimated and the Hausman Test was used to identify the preferred model.

$$SRI_{it} = \beta_0 + \beta_1 endline_t + \vec{\beta}_2 \vec{X}_{it} + \beta_3 A_i + u_{it} \quad (2)$$

This secondary analysis was deemed exempt from review by the Institutional Review Board at Washington University in St. Louis.

### Results

#### Baseline characteristics

Participants who lost to follow-up did not exhibit statistically significant differences in key characteristics or self-reliance as compared to those who were measured both before and after the intervention. Table 1 presents demographic and housing data for a sample of 517 individuals at baseline. Most respondents (506, or 97.9%) were Venezuelan migrants, distributed across four central

**Table 1.** Demographics and other covariates at baseline (*N* = 517)

| Study covariates | Total, *N* = 517 |
|---|---|
| **Region** | |
| Antioquia | 103 (19.9%) |
| Caribbean | 144 (27.9%) |
| Cauca valley | 127 (24.6%) |
| North of Santander | 143 (27.7%) |
| **Household profile** | |
| Venezuelan migrants | 506 (97.9%) |
| Colombian-Venezuelans | 11 (2.1%) |
| **Head of household** | |
| Yes | 514 (99.4%) |
| No | 3 (0.6%) |
| **Nationality** | |
| Venezuelan | 487 (94.2%) |
| Dual nationality-Venezuelan/Colombian | 30 (5.8%) |
| Household size, including the interviewee: Mean (SD) | 4.26 (1.44) |
| **Sex** | |
| Female | 516 (99.8%) |
| Male | 1 (0.2%) |
| Age | 35.88 [11.00] |
| **Breastfeeding** | |
| Yes | 79 (95.2%) |
| No | 4 (4.8%) |
| **Type of housing** | |
| House | 273 (52.8%) |
| Apartment | 186 (36.0%) |
| Room(s) | 48 (9.3%) |
| Other housing (tent, wagon, boat, natural shelter, etc.) | 10 (1.9%) |

*Note*: Statistics are *n*(%) or mean[SD].

regions: Caribbean (144, 27.9%), North of Santander (143, 27.7%), Cauca Valley (127, 24.6%) and Antioquia (103, 19.9%). Nearly all respondents – (514, 99.4%) – reported being the head of their family, with an average household size of four people. Almost all respondents were female (516, 99.8%), and the average respondent was 35.88 years of age. The majority of participants lived in a house (273, 52.8%) or apartment (186, 36.0%), and only 10 respondents (1.9%) were in other forms of shelter, such as tents or boats.

#### Self-reliance index

OLS estimates revealed positive average changes over time for all outcomes, after controlling for observable household and respondent characteristics. Significant improvements were observed from baseline to endline across most SRI domains, including rent, food, education, healthcare, safety, financial resources, debt, savings, financial capital and relational capital, with all scores showing increases at endline at *p*-values below 0.001 (see Table 2). The

**Table 2.** Changes in time after controlling for observed household and respondent characteristics, OLS regressions

| | *B*<br>[95% CI] |
|---|---|
| Housing | 0.00<br>[0.00,0.00] |
| Rent | 2.03***<br>[1.85,2.21] |
| Food | 0.92***<br>[0.82,1.02] |
| Education | 0.43***<br>[0.29,0.57] |
| Health care | 0.62***<br>[0.47,0.77] |
| Health status | 0.30***<br>[0.15,0.45] |
| Safety | 0.79***<br>[0.657,0.929] |
| Housing | 1.11***<br>[0.990,1.239] |
| Financial resources | 0.37***<br>[0.237,0.506] |
| Assistance | −1.51***<br>[−1.608,1.413] |
| Debt | 1.05***<br>[0.910,1.183] |
| Savings | 2.35***<br>[2.232,2.468] |
| Financial capital | 1.24***<br>[1.045,1.443] |
| Relational capital | 0.89***<br>[0.740,1.047] |
| Overall SRI score | 0.65***<br>[0.580,0.711] |
| Subjective well-being | 1.50***<br>[1.395,1.614] |
| Perceived agency | 1.47***<br>[1.356,1.577] |
| Housing has adequate space | 1.99***<br>[1.873,2.104] |
| Housing is safe | 1.52***<br>[1.425,1.619] |
| Housing has adequate privacy | 1.77***<br>[1.657,1.883] |
| Housing is protected from the elements | 1.47***<br>[1.375,1.573] |

*Note*: Models control for point of data collection, respondent's age, respondent's disability status and household size. *p*-values are significant at *$p < 0.05$, **$p < 0.01$ and ***$p < 0.001$.

overall SRI score increased by $B = 0.65$ (95% confidence interval [CI] = [0.58,0.71]). Given that the baseline SRI measure was collected immediately after participants moved to their new and adequate shelters, the housing domain score remained unchanged at 5.00. Subjective well-being and perceived agency both improved significantly by $B = 1.50$ (95% CI = [1.40,1.61]) and $B = 1.47$ (95% CI = [1.36,1.58]), respectively. All four indicators of housing conditions also improved at statistically significant levels ($p < 0.001$).

### Association between self-reliance and potential covariates

Table 3 presents findings from six OLS and six fixed-effects models, where each model controls for the time of data collection and one of the covariates of interest (subjective well-being, perceived agency and each of the four housing indicators).[1] In both the OLS regression models (column 1) and the fixed-effects models (column 2), the endline effect remains statistically significant even when controlling for any of the six covariates, providing stronger evidence in favor of a positive change before and after the housing assistance program was provided. Each covariate is also found to be associated with self-reliance in the OLS regression models. For example, a per-unit increase in subjective well-being is associated with a $B = 0.12$ increase (95% CI = [0.081,0.154]) in self-reliance. However, when controlling for observed and unobserved time-invariant characteristics (column 2), changes in housing with adequate space or adequate privacy are no longer associated with changes in self-reliance.

### Discussion

This study presents findings from a pre-post evaluation of a rental assistance program for female-headed Venezuelan migrant households in Colombia. Analysis revealed a statistically significant increase in overall household-level self-reliance, subjective well-being, perceived agency and housing outcomes. Improvements were seen for all underlying domains of self-reliance, including those that reflect longer-term investment in households' financial stability, such as employment and savings. The intervention was statistically significantly associated with improved self-reliance, even after controlling for observed and unobserved time-invariant characteristics. Finally, after controlling for these time-invariant factors, subjective well-being, perceived agency, having safe housing and having housing protected from weather events were also found to be associated with self-reliance.

Findings from this study offer empirical support for the underlying theory of change guiding the intervention – namely, that housing assistance may foster improvements in household self-reliance, perceived agency and subjective well-being through both material and psychological pathways. Improvements in nearly all domains of the SRI indicate that the intervention had broad and multidimensional effects. While increases in domains such as food security and healthcare access may be explained by an increase in cash flow – freed up by the removal of rental costs – improvements in more future-oriented domains like employment, savings and financial resources likely reflect enhanced cognitive and mental capacity to invest in long-term goals. These findings align with the Scarcity Mindset Framework – which posits that material insecurity depletes mental bandwidth and inhibits long-term planning and goal-directed behavior (de Bruijn and Antonides, 2022) – and the improvements observed for perceived agency further support this interpretation. Although the directionality of the relationship between self-reliance and agency is not yet well established in the literature, it is plausible that housing stability and financial relief helped participants feel more in control of their circumstances and capable of shaping their futures. This sense of control may, in turn, have reinforced behaviors aligned with greater self-reliance, such as job-seeking and saving.

---

[1]Hausman tests confirmed that fixed-effect models were a better fit of the data to account for unobserved and observed time-invariant factors.

**Table 3.** OLS and fixed effects models estimating self-reliance

|  | OLS models<br>*B* [95% CI] | Fixed-effects models<br>*B* [95% CI] |
| --- | --- | --- |
| Endline | 0.47***<br>[0.38,0.55] | 0.54***<br>[0.45,0.63] |
| Subjective well-being | 0.12***<br>[0.081,0.154] | 0.07**<br>[0.024,0.116] |
| Endline | 0.54***<br>[0.46,0.63] | 0.56***<br>[0.476,0.651] |
| Perceived agency | 0.07***<br>[0.035,0.108] | 0.06*<br>[0.011, 0.101] |
| Endline | 0.55***<br>[0.45,0.64] | 0.60***<br>[0.500,0.709] |
| Housing has adequate space | 0.05**<br>[0.015,0.085] | 0.02<br>[−0.023, 0.065] |
| Endline | 0.48***<br>[0.389,0.571] | 0.47***<br>[0.377,0.571] |
| Housing is safe | 0.11***<br>[0.068,0.151] | 0.11***<br>[0.062,0.166] |
| Endline | 0.56***<br>[0.47,0.66] | 0.60***<br>[0.504,0.698] |
| Housing has adequate privacy | 0.05*<br>[0.010,0.081] | 0.03<br>[−0.018,0.069] |
| Endline | 0.54***<br>[0.45,0.63] | 0.51***<br>[0.416,0.602] |
| Housing protects against the elements | 0.07***<br>[0.034,0.115] | 0.09***<br>[0.043,0.143] |
| Overall endline effect | 0.65***<br>[0.579,0.712] | 0.65***<br>[0.588,0.703] |

*Note*: *p*-values are significant at *$p < 0.05$, **$p < 0.01$ and ***$p < 0.001$.

Although improvements in self-reliance and agency may, in turn, have bolstered subjective well-being, this outcome may also have been impacted more directly by the intervention itself, whereby participants experienced reduced anxiety and distress as a result of not having to cover their rent. This hypothesis aligns with prior research from high-income contexts, which shows that rent-related anxiety contributes to psychological distress among low-income households (Denary et al., 2021). Further, participants in this study were required to identify and move to housing that met specific criteria around safety, privacy and protection – housing conditions that few had access to prior to the intervention – in order to qualify for rental support. In the Colombian context, these housing standards were operationalized as follows: adequate space required sufficient room for household members without excessive overcrowding (a common issue in informal settlements); privacy meant separate sleeping areas and basic amenities not shared with other households; safety included secure doors and windows, structural integrity and location in neighborhoods without high rates of violence; and protection from the elements required intact roofing, walls and windows that could withstand Colombia's varied weather conditions, including heavy rains. Recent evidence from high-income contexts has shown that refugees in overcrowded houses, poor-quality housing or housing with insecure tenure faced greater risks of mental illness (Brown et al., 2024; Rana et al., 2025). Because this study employed only two time points, it is not possible to disentangle whether the intervention led to independent gains in

self-reliance, agency and well-being or whether improvements in one domain mediated change in the others. Future studies incorporating longitudinal data with more points of data collection and mixed methods are needed to better unpack these dynamic and potentially recursive pathways.

Study findings also offer important insights for the design and implementation of housing interventions targeting refugee populations in LMIC settings. While all four indicators of housing conditions – adequate space, privacy, safety and protection from the elements – improved from baseline to endline (namely because receiving rental support was conditional on finding housing with these characteristics), only safety and protection from the weather showed statistically significant associations with improvements in household self-reliance. This finding suggests that not all dimensions of housing quality contribute equally to self-reliance and well-being, and that program implementers may want to prioritize supporting access to safe and secure housing when housing with all four qualities is scarce. In the context of Colombia, housing-related safety may play an especially critical role in overall perceived safety – for example, the average SRI safety score increased from 4.56 at baseline to 4.99 at endline, and only one respondent at endline reported feeling unsafe to the extent that it prevented them from pursuing any opportunities. In addition, while short-term rental assistance clearly improved housing and self-reliance in the study period, the optimal duration of such support and the sustainability of outcomes post-intervention remain open questions.

Future research is needed to build the evidence base on housing interventions for refugees in LMIC contexts. Subsequent evaluations should also measure outcomes further out from endline in order to examine the sustainability of impacts. For example, such an approach would enable assessment of whether participants are able to maintain their newly achieved self-reliance once they resume responsibility for rental payments. Importantly, future evaluations of this specific rental assistance program must incorporate more robust designs – such as control groups and random assignment – in order to more confidently attribute changes in outcomes of interest to the intervention.

This study includes several limitations of note. First, the evaluation did not include a comparison or control group, limiting the ability to attribute observed changes over time to the intervention itself. Second, perceived agency and subjective well-being were each measured using only one question and were both treated as continuous in regression models despite being ordinal. Agency and well-being are complex constructs, and future research should incorporate more robust, validated measures of these outcomes. Third, 16% of participants were lost to follow-up, which may have introduced bias if those who discontinued differed from those retained with respect to self-reliance or other unobserved characteristics. It is also important to note that this implementation of the intervention targeted a specific subgroup of forcibly displaced populations: female-headed households. Future research is needed to examine the utility of housing support for other vulnerable, forcibly displaced groups. Finally, this study did not assess whether or how the intervention may have interacted with the broader community, potentially impacting refugees' integration outcomes and/or their neighbors' attitudes toward refugees. For example, a recent study from Jordan, which evaluated a housing subsidy program for Syrian refugees, reported increased tensions between the refugee participants and host communities, suggesting that housing assistance could inadvertently affect host communities' perceptions of fairness and social cohesion (Tamim et al., 2025). Although such effects were not measured in the present evaluation,

they remain important considerations for future programming in Colombia and similar settings, where large-scale migration continues to reshape urban neighborhoods.

## Conclusion

This study contributes to a growing but still limited body of research examining the effects of housing interventions on refugee outcomes in LMIC contexts. Findings lend empirical support for the theory that housing assistance may improve household self-reliance and psychosocial outcomes not only by alleviating financial distress, but also by freeing cognitive and emotional bandwidth necessary for long-term planning and goal-directed behavior. However, only select housing characteristics, notably safety and protection from the elements, were associated with increased self-reliance, suggesting that enhancing these specific features may be especially important for promoting sustainable outcomes for displaced populations.

Future evaluations should incorporate longer-term follow-up and additional time points to better capture the trajectory and sustainability of outcomes. Moreover, a more rigorous design – including control groups and randomized assignment – is needed to establish causal relationships and to explore potential externalities, including effects on community integration and host perceptions. Ultimately, as housing insecurity continues to shape the lives of refugees and migrants in urban LMIC contexts, findings from this study offer valuable insights for policymakers and practitioners working to design more effective, equitable and sustainable housing programs for displaced populations.

**Open peer review.** To view the open peer review materials for this article, please visit http://doi.org/10.1017/gmh.2025.10106.

**Data availability statement.** At the moment, the data sharing agreement with the organization only extends to the authors. Authors are in discussions about creating a de-identified data collection that can be shared openly.

**Acknowledgments.** The authors would like to acknowledge all of the participants who devoted time to participating in Blumont's rental assistance program.

**Author contribution.** Conceptualization:- IS, LS, AHR, JPF, NM and DB. Methodology: IS, LS, AHR and JPF. Data curation: IS and NM. Data analysis: IS and NM. Writing – original draft: IS and LS. Writing – reviewing and editing: All authors.

**Financial support.** This study was made possible by the support of the American people through the US State Department Bureau of Population, Refugees and Migration. The findings of this study are the sole responsibility of the contributing authors and do not necessarily reflect the views of the US government. This research was funded by Washington University in St Louis, under grant numbers **PJ000032416** and **PJ000032066.**

**Competing interests.** The authors declare none.

**Ethics statement.** This secondary analysis was deemed exempt from review by the Institutional Review Board at Washington University in St. Louis.

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
