## [Reviewer Report]

This study provides preliminary evidence that rental assistance can enhance both material and psychosocial well-being among displaced populations in low- and middle-income countries. Focusing on a nine-month rental support program for female-headed Venezuelan migrant households in Colombia, the authors found significant gains in self-reliance, well-being, and perceived agency. A key contribution of the study is its demonstration that housing interventions not only improve economic stability but also foster psychological recovery and a sense of agency. However, I believe the manuscript would benefit from the following revisions:

1. In the first paragraph of the introduction, the manuscript states: “As of 2023, there were 50.3 refugees, asylum seekers…”. Please clarify whether this figure refers to 50.3 million individuals, as the current phrasing is ambiguous.

2. The article investigates the impact of a rental assistance program on mental health outcomes among refugees. While there may be limited research specifically focused on refugees, there is a substantial body of literature examining the effects of rental assistance on mental health in other populations. Integrating this broader research would strengthen the introduction and better situate the study within the existing literature.

3. In the “Participants and Procedures” section, it is noted that 16% of the sample was lost to follow-up. This attrition should be acknowledged and discussed in the limitations section, as the characteristics of those lost to follow-up may systematically differ from those retained, potentially biasing the findings.

4. The primary outcome (SRI) was measured after participants had moved into new accommodations, although the questions refer to experiences over the past 12 months. The authors should discuss whether post-move measurement may positively bias participants’ self-reported self-reliance and consider this as a potential limitation.

5. In the “Analysis” section, the authors refer to “treatment effects” on the SRI. Given the absence of a control group, this terminology may be misleading. Consider using alternative language such as “program effects” or “associations” to more accurately reflect the study design.

6. In Table 2, the coefficient for Housing is reported as 0. For consistency and clarity, please format this to match the decimal places used for other coefficients (e.g., 0.00).

7. In Table 3, the “Overall endline effect” for the OLS models is reported with three decimal places, while other coefficients use two. Please standardize the decimal formatting throughout the table for consistency.

8. The Discussion section references an underlying theory of change; however, this framework is not introduced earlier in the manuscript. Consider including a brief discussion of the theory of change in the Introduction to better contextualize the study’s rationale and hypotheses.

---

## [Reviewer Report]

The main merits of this article are its focus on a topic of much needed study, and the use of pertinent bibliography and adequate statistical management. The main comments/suggestions of this reviewer follow: a) To emphasize the need to produce more studies, particularly centered on representativeness of the samples, a missing feature here; b) Related to (a), it is necessary to speculate about the circumstances of the study survey, and the impact of personal/group convenience factors in the provision of responses, how to methodologically prevent them, etc.; c) Introduction Section, p. 3, l. 2: the phrase “50.3 refugees, asylum seekers and other forced migrants in need of....” is certainly incomplete; please, correct it; d) Results section, Baseline Characteristics subsection, p. 8, lines 8 and 9: The phrase “...a majority of participants (273, 52,8%)....” is repeated; please, correct; e) Discussion section: Some details are needed to clarify and precise the “housing conditions” in LMIC settings, as mentioned; f) Conclusion section: Par 1, lines 5-9 include redundant statements about the concept of self-reliance, and leave unclear the “certain aspects of housing quality” that may require “the promotion of sustainable outcomes”.

---

## [Editor Report]

Dear researcher, We have received comments from the reviewers on your work, and both agree that it is a valuable contribution, but it requires some adjustments to the manuscript, which I detail below.

Reviewer 1:

The main merits of this article are its focus on a topic of much needed study, and the use of pertinent bibliography and adequate statistical management. The main comments/suggestions of this reviewer follow: a) To emphasize the need to produce more studies, particularly centered on representativeness of the samples, a missing feature here; b) Related to (a), it is necessary to speculate about the circumstances of the study survey, and the impact of personal/group convenience factors in the provision of responses, how to methodologically prevent them, etc.; c) Introduction Section, p. 3, l. 2: the phrase “50.3 refugees, asylum seekers and other forced migrants in need of....” is certainly incomplete; please, correct it; d) Results section, Baseline Characteristics subsection, p. 8, lines 8 and 9: The phrase “...a majority of participants (273, 52,8%)....” is repeated; please, correct; e) Discussion section: Some details are needed to clarify and precise the “housing conditions” in LMIC settings, as mentioned; f) Conclusion section: Par 1, lines 5-9 include redundant statements about the concept of self-reliance, and leave unclear the “certain aspects of housing quality” that may require “the promotion of sustainable outcomes”. 

Reviewer 2:

This study provides preliminary evidence that rental assistance can enhance both material and psychosocial well-being among displaced populations in low- and middle-income countries. Focusing on a nine-month rental support program for female-headed Venezuelan migrant households in Colombia, the authors found significant gains in self-reliance, well-being, and perceived agency. A key contribution of the study is its demonstration that housing interventions not only improve economic stability but also foster psychological recovery and a sense of agency. However, I believe the manuscript would benefit from the following revisions:

1. In the first paragraph of the introduction, the manuscript states: “As of 2023, there were 50.3 refugees, asylum seekers…”. Please clarify whether this figure refers to 50.3 million individuals, as the current phrasing is ambiguous.

2. The article investigates the impact of a rental assistance program on mental health outcomes among refugees. While there may be limited research specifically focused on refugees, there is a substantial body of literature examining the effects of rental assistance on mental health in other populations. Integrating this broader research would strengthen the introduction and better situate the study within the existing literature.

3. In the “Participants and Procedures” section, it is noted that 16% of the sample was lost to follow-up. This attrition should be acknowledged and discussed in the limitations section, as the characteristics of those lost to follow-up may systematically differ from those retained, potentially biasing the findings.

4. The primary outcome (SRI) was measured after participants had moved into new accommodations, although the questions refer to experiences over the past 12 months. The authors should discuss whether post-move measurement may positively bias participants’ self-reported self-reliance and consider this as a potential limitation.

5. In the “Analysis” section, the authors refer to “treatment effects” on the SRI. Given the absence of a control group, this terminology may be misleading. Consider using alternative language such as “program effects” or “associations” to more accurately reflect the study design.

6. In Table 2, the coefficient for Housing is reported as 0. For consistency and clarity, please format this to match the decimal places used for other coefficients (e.g., 0.00).

7. In Table 3, the “Overall endline effect” for the OLS models is reported with three decimal places, while other coefficients use two. Please standardize the decimal formatting throughout the table for consistency.

8. The Discussion section references an underlying theory of change; however, this framework is not introduced earlier in the manuscript. Consider including a brief discussion of the theory of change in the Introduction to better contextualize the study’s rationale and hypotheses.

We hope these comments will be useful in improving your manuscript and look forward to a new version.

---

## [Reviewer Report]

Very well-conceived and structured presentation of a study on an important connection between social factors and their impact on mental health of Venezuelan migrants in Colombia. Very clear description of the target population, the housing process, measurement procedures, provision and description of results. Use of sophisticated methodological procedures. Objective assessment of advantages and limitations of the study, potential clinical impact of housing insecurity, findings/results sustainability features, and delineation of longitudinal designs and control groups as needed features in future assessments of refugee/migrant populations.

---

## [Editor Report]

Dear authors,

Your manuscript has completed peer review and has been accepted for publication in our journal.

Congratulations!

Best regards.